# Bacterial Adhesion Capacity of Uropathogenic *Escherichia coli* to Polyelectrolyte Multilayer Coated Urinary Catheter Surface

**Klemen Bohinc** [1,*], **Lora Kukić** [2], **Roman Štukelj** [1], **Anamarija Zore** [1], **Anže Abram** [3], **Tin Klačić** [4] **and Davor Kovačević** [4]

1 Faculty of Health Sciences, University of Ljubljana, Zdravstvena pot 5, 1000 Ljubljana, Slovenia; stukeljr@zf.uni-lj.si (R.Š.); anamarija.zore@zf.uni-lj.si (A.Z.)
2 Faculty of Medicine, University of Rijeka, 51000 Rijeka, Croatia; lora.kukic@gmail.com
3 Department for Nanostructured Materials, Jožef Stefan Institute, 1000 Ljubljana, Slovenia; anze.abram@ijs.si
4 Department of Chemistry, Faculty of Science, University of Zagreb, 10000 Zagreb, Croatia; tklacic@chem.pmf.hr (T.K.); davork@chem.pmf.hr (D.K.)
* Correspondence: klemen.bohinc@zf.uni-lj.si

**Abstract:** The application of catheters to the urinary tract is associated with nosocomial infections. Such infections are one of the most common types of infections in hospitals and health care facilities and can lead to numerous medical complications. Therefore, the understanding of the properties of urinary catheter surfaces and their potential modifications are crucial in order to reduce bacterial adhesion and subsequent biofilm formation. In our study, we consider standard polyvinyl chloride (PVC) catheter surfaces and compare their properties with the properties of the same surfaces coated with poly(diallyldimethylammonium chloride)/poly(sodium 4-styrenesulfonate) (PDADMA/PSS) polyelectrolyte multilayers. Uncoated and coated surfaces were characterized by means of roughness, hydrophobicity, and zeta potential measurements. Finally, bacterial adhesion extent of uropathogenic *Escherichia coli* on bare and polyelectrolyte multilayer coated surfaces was measured. The obtained results show that on non-treated surfaces, biofilm is formed which was not the case for multilayer coated surfaces. The PSS-terminated multilayer shows the lowest bacterial adhesion and could be helpful in prevention of biofilm formation. The analysis of the properties of the uncoated and coated surfaces reveals that the most significant difference is related to the charge (i.e., zeta potential) of the examined surfaces, while roughness and hydrophobicity of the examined surfaces are similar. Therefore, it could be concluded that the surface charge plays the crucial role in the bacterial adhesion on uncoated and coated PVC catheter surfaces.

**Keywords:** urinary catheters; polyelectrolyte multilayers; bacterial adhesion; *Escherichia coli*

## 1. Introduction

Each year, there are millions of catheters used worldwide for different indications both diagnostic and therapeutic. In the latter case, the catheter is associated with the urinary tract where the infection caused by patient's microflora can take place. The incidence of bacteriuria (the presence of bacteria in urine) in medical facilities rises by 3%–8% per day for each day after catheter insertion [1]. Most common microorganisms involved in urinary tract infections are *Escherichia coli* (*E. coli*), *Pseudomonas* spp., and *Candida* spp., as well as some other Gram-negative bacterial species from the family of *Enteriobacteriae*.

Bacteria can invade the usually sterile medium extraluminally or intraluminally. The extraluminal way involves the creation of a biofilm, while intraluminal occurs due to stasis in urine. The formation of the biofilm is not only significant as a pathway to an infection, but also often causes the recurrence of infection. The cornerstone of the biofilm formation is bacterial adhesion, which is a complicated process that includes physical and chemical properties of both bacteria and the surface colonies of bacteria. The factors associated with the bacterial adhesion are characteristic properties of bacteria and material

surface. Bacterial properties are given by surface charge and hydrophobicity, as well as by the presence of flagella, bacterial motility, and the production of urease [2]. Studies have shown that the following bacterial strains significantly contribute to the production of biofilms on the catheter surfaces: *E. coli* (60%–80%), *Klebsiella* spp. (16%–21%), coagulase-negative *Staphylococcus* (3%–6%), *Proteus mirabilis* (6%), *Pseudomonas aeruginosa* (2%–4%), *Acinetobacter* spp. (1%–2%), and *Enterococcus* (2%) [2].

*E. coli* is one the most versatile bacteria found as a member of normal intestinal flora of mammals. Some strains of *E. coli* can be, and are, lethal pathogens [3] and a common cause of enteritis, urinary tract infection, septicaemia, and other clinical infections, such as neonatal meningitis. With antimicrobial resistance on the rise, the emergence and diffusion of multi-drug resistant strains of *E. coli* causes complications in the treatment of several serious infections. *E. coli* is the most frequent cause of hospital and community-acquired infections [4].

Material surface characteristics crucial for the bacterial adhesion are the material surface energy, roughness, wettability, and zeta potential [5]. These characteristics are accessible by techniques like profilometry, atomic force microscopy, tensiometry, and electrophoresis. Advanced techniques make it possible to gain new knowledge about the bacterial adhesion and subsequent colonization of microorganisms to different type of surfaces [6].

Medical products can be improved by surface modifications. This also applies to catheter surfaces, for which the bacterial adhesion needs to be minimized and subsequent biofilm formation needs to be suppressed. For this purpose, various coatings could be produced. Among coatings, the preparation of the polyelectrolyte multilayer (PEM) based on the layer-by-layer (LbL) technique is very promising [7]. This technique is adaptable also to catheter surfaces and various other types of surfaces [8–10]. The coatings obtained by alternate adsorption of oppositely charged polyelectrolytes usually have a thickness of only a few nanometers. The advantage of such films is that they are robust coatings and could be used for long-lasting surface modification. Additionally, PEMs can be prepared as slowly degradable coatings for controlled release of active substances. The large variety of available polyelectrolytes can enable the formation of coatings with adjustable biocompatible properties [11–13].

Many studied polyelectrolyte multilayers could promote or disrupt bacterial biofilm formation because of their high surface charge density [14,15]. If the terminating layer of PEM has an opposite charge than bacteria, the interaction between bacteria and PEM is promoted. If the PEM terminated layer and bacterial cell surface have a similar charge, then the bacterial adhesion is hindered. In the recent study, we showed that the optimization of poly(allylamine hydrochloride)/sodium poly(4-styrenesulfonate) (PAH/PSS) polyelectrolyte multilayers [14] can lead to antibacterial properties. The option is also to perform PEM with the terminated layer composed of proteins [15]. It was found that the bacterial adhesion on protein layers strongly depends on the protein specificity and the surface physical properties do not play the key role in the adhesion process [15].

The aim of this study was to extend our investigations on PEM-modified surfaces to the standard catheter surfaces coated with polyelectrolyte multilayers. This presents a medically relevant application for the prevention of infections on catheters. For that purpose, we applied zeta potential measurements to determine the charge of the bare catheter surface and catheter surfaces coated with polyelectrolyte multilayers. As oppositely-charged polyelectrolytes, we used poly(diallyldimethylammonium chloride) (PDADMA) and poly(sodium 4-styrenesulfonate) (PSS). These synthetic polyelectrolytes have been widely used in the process of polyelectrolyte multilayer formation [16–18]. The surfaces were additionally characterized by measuring the roughness and hydrophobicity. Finally, the bacterial adhesion extent of uropathogenic *E. coli* on all studied surfaces was determined.

## 2. Materials and Methods

### 2.1. Materials

2.1.1. Substrate

Medical PVC coupons of diameter 3 cm and thickness 7 mm were used. From this PVC, the urethral catheters are made and produced by the company TIK d.o.o., Kobarid, Slovenia. The coupons were purified with 70% ethanol, irradiated with UV light for 30 min each side, and transferred into 6-wells microtiter dishes. Additionally, few PVC coupons were cut to smaller plates (1 cm × 1 cm × 2 mm) for Atomic force microscopy (AFM) and contact angle measurements.

2.1.2. Polyelectrolytes

The cationic polyelectrolyte poly(diallyldimethylammonium chloride) (PDADMA, $M_w < 100,000$ g·mol$^{-1}$) and the anionic polyelectrolyte poly(sodium 4-styrenesulfonate) (PSS, $M_w \approx 70,000$ g·mol$^{-1}$) were purchased from Sigma-Aldrich (St. Louis, MO, USA) and were used without further purification.

2.1.3. Bacteria

*E. coli* ACTT 77115, uropathogenic strains used in our study were cultivated on selective XLD agar for 24 h on 37 °C in aerobic conditions. Next day, the overnight culture was prepared in BHI (brain-heart infusion) nutrient broth (Biolife, Italiana Srl, Milano, Italy) at 37 °C for 18 h to obtain $10^9$ CFU/mL bacterial suspension. The reference strain of positive biofilm producer *E. coli* ATCC 35218 was used as a control.

### 2.2. Methods

2.2.1. Polyelectrolyte Multilayer Preparation

Solutions of PDADMA and PSS were prepared in 0.50 mol/L NaCl (Sigma Aldrich, St. Louis, MO, USA) supporting electrolyte with ultra-pure water (Milli Q Plus system, Millipore, Billerica, MA, USA). The concentration of the respective polyelectrolyte in solutions was 0.01 mol/L (based on monomer repeat unit). To ensure the maximum adsorption of both polyions on the substrate surface, the pH of polyelectrolyte solutions was adjusted to 7.9 ± 0.2 with 1.0 mol/L NaOH solution (Titrisol, Merck, Darmstadt, Germany). For that purpose, pH-meter (826 pH mobile, Metrohm, Herisau, Switzerland) equipped with a combined glass microelectrode (6.0234.100, Metrohm) was used. Prior to measurements, the potentiometric system was calibrated with standard buffers (Sigma-Aldrich (St. Louis, MO, USA) of pH 3.0, 5.0, 7.0, and 9.0.

The polyelectrolyte multilayers were prepared according to the layer-by-layer deposition technique suggested by Decher [19]. The sequential dipping of the substrate in the polyanion and polycation solutions was carried out in the following way. The PVC substrate was affixed to a steel shaft and was immersed for 5 min in 25 mL PSS solution. After deposition, the coated substrate was gently rinsed with enough ultra-pure water and was then blown dry with argon (Messer, Bad Soden, Germany). The described process was repeated for PDADMA solution, constructing a one bilayer of polyelectrolytes on the PVC surface. The PSS and PDADMA adsorptions were continued until the desired number of layers was attained. Two samples were prepared, one with three and one with four layers. For simplicity, the prepared multilayers are designated as (PSS/PDADMA)-PSS when the last layer is negatively charged PSS and (PSS/PDADMA)$_2$ when the last layer is positively charged PDADMA.

2.2.2. Surface Morphology and Roughness

The catheter's surface topography and roughness were determined by the soft tapping mode AFM) using a Multimode 8 apparatus from Bruker (Billerica, MA, USA). All AFM measurements were carried out with NCHV-A probe (Bruker) in ambient air conditions at a temperature of (25 ± 2) °C and relative humidity between 25% and 35%. The used PVC probe has a rectangular cantilever of 117 µm in length and 33 µm in width with a resonance

frequency of 320 kHz and a spring constant of 42 N·m$^{-1}$. The tip height of the probe is 10–15 μm having a nominal radius of 8 nm. The samples were scanned at a speed of 0.5 Hz. The scan size was 50 μm × 50 μm and the image resolution was 512 × 512 pixels. For the roughness determination, surface images were recorded at five different positions on the sample. After the images were corrected for tilt and bow using a second order flatten fit, average values, and standard errors were calculated. For processing the measurements and data analysis, NanoScope 9.7 and NanoScope Analysis 2.0 software packages (Bruker) were used.

### 2.2.3. Zeta Potential Measurements

The zeta potential of the surface was determined by an electro-kinetic analyzer (Sur-PASS, Anton Paar GmbH, Graz, Austria). At room temperature, 1 mM phosphate-buffered saline (PBS) solution was forced to flow through a capillary and the electrical potential was produced between the ends of the capillary. This electrical difference is denoted by the streaming potential. The zeta potential was calculated from the streaming potential using Helmholtz–Smoluchowski equation.

### 2.2.4. Contact Angle Measurements

Contact angle measurements were performed by Attension Theta (Biolin Scientific, Gothenburg, Sweden) tensiometer which includes a light source, camera, liquid dispenser, and a sample stage. Catheter surfaces were placed on the sample stage, a liquid droplet was put on the material surface, and the contact angle between the droplet and the catheter surface was measured. Several measurements were made for each material, from which the average value of the contact angle was calculated.

### 2.2.5. Monitoring of Bacterial Adhesion on Flat Catheter Surfaces

The adhesion of *E. coli* to catheter surfaces was made by the procedure described by Bohinc et al. [20–22] with some modifications. First, we immersed each specimen into the diluted (1:300) overnight culture of *E. coli* with BHI broth. Specimens were incubated for 10 h and afterwards the attached bacteria were fixed with 0.1 M PBS and hot air. At the end, specimens were washed in distilled H$_2$O and hot air-dried.

Bacterial adhesion was detected by scanning electron microscopy (SEM) Jeol JSM-7600F (Tokyo, Japan). A thin gold layer (7 nm) was applied to the catheter surface beforehand to achieve a conductive sample with a GATAN Model 682 PECS system (Precision Ion Etching and Coating System, GATAN Inc., Pleasanton, CA, USA). For quantitative analysis, we used ImageJ software package (Version 1.50b, 2015, Wayne Rasband, National Institutes of Health, Bethesda, MD, USA) to count the adhered bacteria.

### 2.2.6. Monitoring of Bacterial Adhesion on Cylindrical Catheter Surfaces

The fluid flow system applied for studying bacterial adhesion on cylindrical catheter surfaces was composed of the bottle with suspension of E. coli ATCC 35218 and cylindrical catheter connected to the bottle with flow 20 mL/h. The inner part of the catheter was exposed for 3 h, 12 h, and 24 h of laminar liquid flow. After the experiment, the inner part of the catheter was examined with SEM. Bacterial adhesion was detected by scanning electron microscopy (SEM) Jeol JSM-7600F (Tokyo, Japan) as explained in the Section 2.2.5.

### 2.2.7. Statistical Analysis

We used the MATLAB software (version 9.0) to perform the statistical analysis. The results were compared by Student's *t*-test at 5% probability level.

## 3. Results

### *3.1. Surface Topography and Roughness*

As mentioned in the Introduction, bacterial adhesion is affected by the surface properties of the substrate. One of these properties is surface morphology which is in close

correlation with surface roughness. Figure 1A shows the surface morphology of PVC catheters. One can see that the PVC surface is rough and non-uniform with slightly pronounced line patterns. These line patterns are probably a consequence of the method used by the manufacturer for PVC coupons preparation. If we compare height fluctuation on PVC surface and size of bacteria, we may conclude that for bacteria with an average diameter of 2 μm, the surface of urethral catheter looks almost flat.

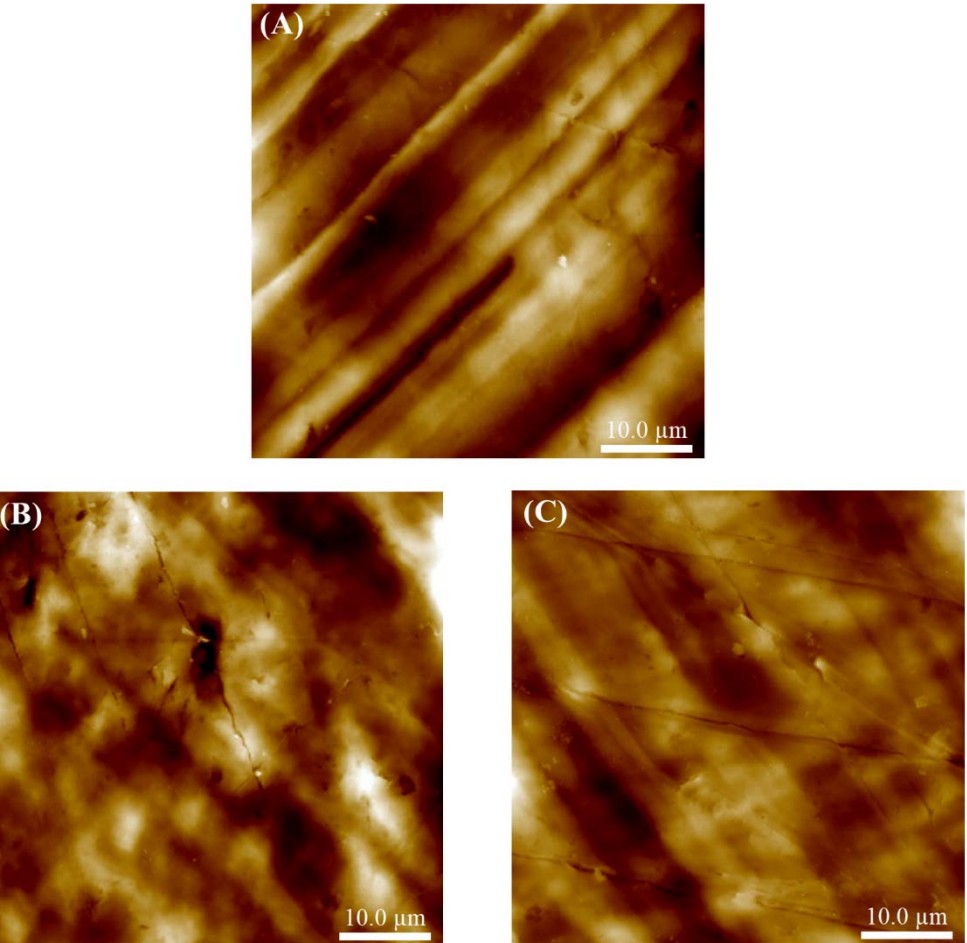

**Figure 1.** AFM images of PVC substrate surface (**A**), surface of PVC substrate coated with PSS-terminated polyelectrolyte multilayer (**B**), and surface of PVC substrate coated with PDADMA-terminated polyelectrolyte multilayer (**C**). On each AFM image z-scale is 1.2 μm.

After we modified the PVC surface by the LbL technique, morphological changes were made. Figure 1B,C represents the PVC surface fabricated with three and four PSS/PDADMA layers, respectively. Although the surface of catheter material covered with PSS/PDADMA assembly is also rough and non-uniform like bare substrate, line patterns are no more visible on the surfaces. Instead of line patterns, furrows of nanometer depth have appeared after the fabrication process. One may state that these surface defects are cracks in the LbL film structure. An interesting fact about these defects is that they were found on both prepared polymer surfaces. Additionally, it is worth mentioning that there are negligible morphological differences between multilayer terminated with PSS and PDADMA layer. This founding reflects the small difference in the surface roughness of these two samples. Roughness parameters of the bare catheter surface and surface covered with polyelectrolyte multilayers are shown in Table 1 and Figure 2. In general, all three examined surfaces have similar $R_a$ and $R_q$ values. This means that they are of similar roughness and thus of similar surface topography. A result like this is not so unexpected

because bare PVC substrate has a rough surface ($R_q$ = 160 nm) and formation of nanofilm on its surface makes only small changes in roughness as polymer material clings to the surface of the substrate in a very thin layer. The statistical analysis shows that the roughness of PVC is not significantly different from the roughness of PSS- ($p$ = 0.11) and PDADMA- ($p$ = 0.08) terminated multilayer.

**Table 1.** Quantitative measurements of the local average surface roughness ($R_a$) and root-mean-square (RMS) surface roughness ($R_q$) made on the PVC catheter sample and the same material covered with PSS/PDADMA films.

| Sample | $R_a$/nm | $R_q$/nm |
|---|---|---|
| PVC substrate | 128 ± 6 | 160 ± 5 |
| PVC substrate coated with PSS-terminated multilayer | 141 ± 14 | 179 ± 18 |
| PVC substrate coated with PDADMA-terminated multilayer | 119 ± 6 | 149 ± 7 |

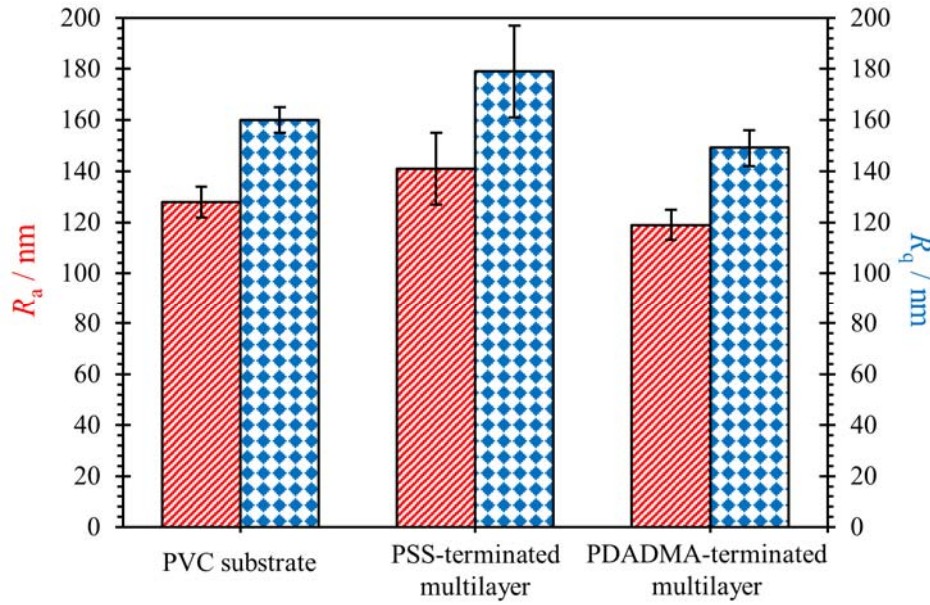

**Figure 2.** Quantitative measurements of the local average surface roughness ($R_a$) and root-mean-square (RMS) surface roughness ($R_q$) made on the PVC catheter sample and the same material covered with PSS/PDADMA films.

*3.2. Zeta Potential*

The zeta potential measurements (Table 2) indicated that the bar PVC catheter surface is positively charged with the zeta potential (15.84 ± 2) mV. The PEM coating with the PSS-terminated layer is negatively charged (−63.25 ± 0.35) mV, whereas the PEM coating with the PDADMA-terminated layer is positively charged (4.19 ± 0.25) mV.

**Table 2.** Zeta potential of uncoated and coated PVC surfaces at pH = 7.7.

| Sample | ($\zeta$ + SE)/mV |
|---|---|
| PVC substrate | +15.84 ± 2.0 |
| PVC substrate coated with PSS-terminated multilayer | −63.25 ± 0.35 |
| PVC substrate coated with PDADMA-terminated multilayer | +4.19 ± 0.25 |

$\zeta$ is the zeta potential, SE is the standard error.

*3.3. Contact Angle Measurements*

The advanced and receding contact angles of a water droplet on different material surfaces were measured with the optical tensiometer (Figure 3). For each surface type,

three measurements were performed and the average contact angle with its standard deviation was calculated. Generally, surfaces with contact angles larger than 90° are considered hydrophobic, whereas surfaces with contact angles smaller than 90° are treated as hydrophilic.

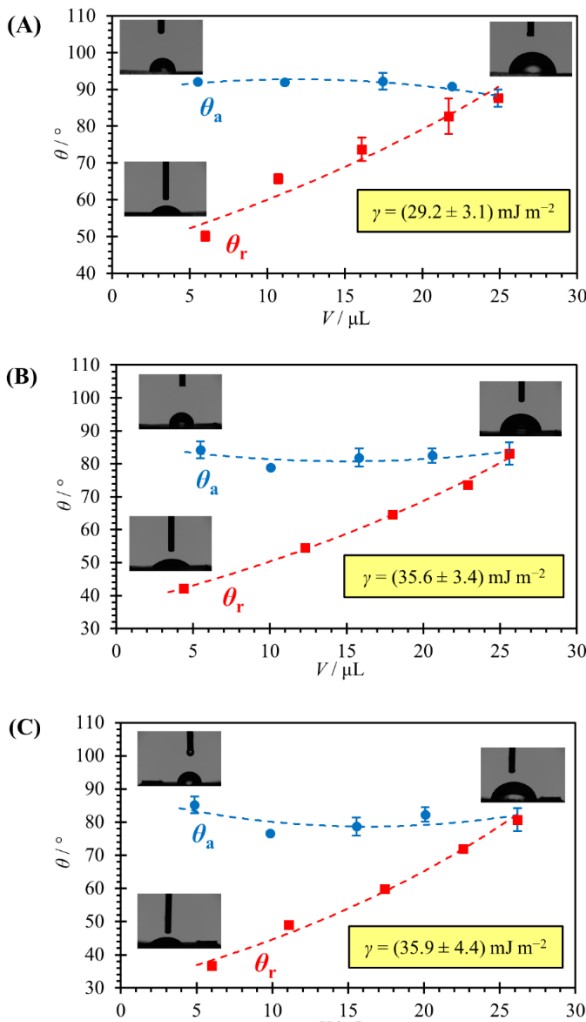

**Figure 3.** Advanced and receding contact angles as a function of the droplet volume. Three different cases are considered: PVC (**A**), PSS-terminated multilayer (**B**), and PDADMA-terminated multilayer (**C**). The insets show the liquid droplet on the corresponding surfaces. The calculated surface free energy γ values are also included.

The static angles can be estimated at larger droplet volumes. The largest static contact angle was observed on PVC (91.2° ± 1.3°). The contact angles on the PSS-terminated multilayer and PDADMA-terminated multilayer are lower than on PVC. The static contact angle on the PSS-terminated layer is 82.8° ± 1.6°, whereas on the PDADMA-terminated layer, the contact angle is 83.8° ± 2.0°. Similarly, the lowest surface free energy is for PVC 29.2 mJm$^{-2}$. For the PSS- and PDADMA-terminated multilayer, the surface energies are very similar, 35.6 and 35.9 mJm$^{-2}$, respectively. A *t*-test analysis showed that the surface energy of the PDADMA-terminated multilayer is not significantly different from the PSS-terminated multilayer ($p = 0.17$). The surface energies of the PDADMA-terminated multilayer and PSS-terminated multilayer are significantly different from PVC ($p = 0.0005$).

### 3.4. Bacterial Adhesion Extent

The SEM microscopy was used to image the surfaces of the samples and to determine where the bacteria were adhered. Figure 4 shows micrographs of PVC catheter surfaces (bare and with PEM coating) with attached *E. coli* ACTT 77115 bacteria. The images were made after 10 h of incubation. SEM micrographs of uncoated catheter surfaces show a very compact biofilm adhered to the surface with some ropes and chimneys for the food supply. On contrary, surfaces with PEM coatings have less adhered *E. coli*. Negatively charged PEM with PSS-terminated layers show on average 31 bacteria per surface area 1700 $\mu m^2$, whereas positively charged PEM with the PDADMA-terminated layer includes 50 bacteria per 1700 $\mu m^2$ (Table 3). A *t*-test analysis showed that the number of adhered bacteria per surface area on PEM with the PDADMA-terminated layer was significantly smaller than the number of adhered bacteria on PEM with the PSS-terminated layers ($p = 0.017$).

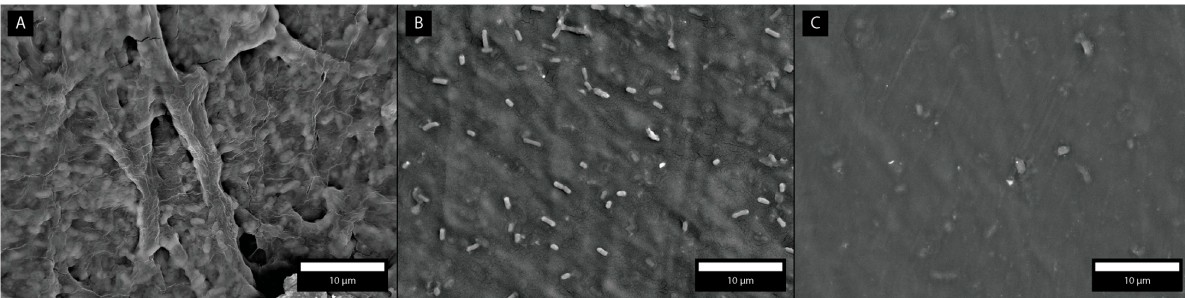

**Figure 4.** SEM micrographs of three different types of surfaces with adhered bacteria *E. coli* ACTT 77115: PVC (**A**) substrate, PVC substrate coated with PSS-terminated multilayer (**B**), and PVC substrate coated with PDADMA-terminated multilayer (**C**).

**Table 3.** Number of adhered bacteria per 1700 $\mu m^2$.

|  |  | PVC Substrate Coated with PDADMA-Terminated Multilayer | PVC Substrate Coated with PSS-Terminated Multilayer |
|---|---|---|---|
|  | PVC Substrate |  |  |
| Number of adhered bacteria per 1700 $\mu m^2$ | Biofilm formation | 50 ± 12 | 31 ± 17 |

### 3.5. Bacterial Adhesion on PVC Catheter under a Liquid Flow

The SEM microscopy was also used to image the inner surfaces of the catheter in the cylindrical geometry. Figure 5 shows micrographs of small parts of the cylindrical surface with adhered *E. coli* ACTT 35218. The images were made after 3h, 12 h, and 24 h of incubation. At shorter incubation time, SEM micrographs show single bacteria attached to the PVC surface. With the increasing incubation time, more bacteria are attached to the surface and after 12 h the biofilm starts to grow.

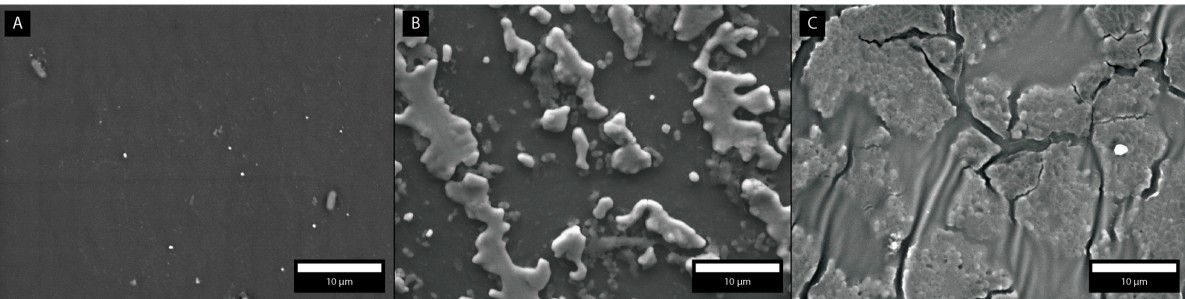

**Figure 5.** SEM micrographs of PVC catheter inner surface in cylindrical geometry with adhered bacteria. *E. coli* ACTT 35218 was exposed to PVC for (**A**) 3 h, (**B**) 12 h, and (**C**) 24 h.

## 4. Discussion

As stated in the Introduction, the aim of this study was to extend our investigations on polyelectrolyte multilayer-modified surfaces [14,15] to the standard PVC catheter surfaces coated with PDADMA/PSS multilayers [17] in order to find a way to reduce bacterial adhesion on catheter surfaces. For that purpose, we use uropathogenic *E. coli* as the bacteria of interest. We were motivated by the well-known fact that nosocomial infections are very frequent in hospitals and health care facilities and can lead to numerous medical complications from mild catheter encrustation and bladder stones to severe septicaemia, endotoxic shock, and pyelonephritis. On the other hand, these infections are also related to a great financial burden to the health care system. There are several strategies that could be used for combating catheter-related complications, one of them being surface modifications [23,24].

It is known [25–28] that the substrate to which bacteria possibly adhere should be characterized in detail, especially in terms of its charge, hydrophobicity, and roughness. Therefore, in the first part of our study, we examined the above-mentioned properties of a bare, uncoated PVC catheter surface and compared the obtained results with the surface properties determined for the same substrate coated with PDADMA/PSS polyelectrolyte multilayers. Polyelectrolyte multilayers terminating with two different polyelectrolytes: a polycation (PDADMA) and a polyanion (PSS) were studied.

Roughness measurements revealed that all three examined surfaces (bare PVC surface and PEM coated surfaces) have similar roughness. It is not surprising because bare PVC substrate has a relatively rough surface ($R_q$ = 160 nm), and therefore the formation of a multilayer does not change that parameter significantly. Similar holds for the measured contact angle (i.e., hydrophobicity). The results show that all examined surfaces are weakly hydrophilic with the contact angle between 80° and 90°. From the results follows that neither roughness not hydrophobicity are the leading factors for bacterial adhesion. On the contrary, the results obtained using streaming potential measurements, showing significant difference for the three studied types of surface. The bare PVC surface is positively charged (zeta potential = (15.84 ± 2) mV). As expected, the PEM coating with the PSS-terminated layer is highly negatively charged ((−63.25 ± 0.35) mV), whereas the PEM coating with the PDADMA-terminated layer is positively charged ((4.19 ± 0.25) mV).

The obtained results allowed us to predict that that negatively charged bacteria [21] would preferably adhere to a bare PVC surface and not so pronounced, on a PDADMA-terminated multilayer. That prediction is in accordance with the results presented in Figure 4 and Table 3. It seems that in the case of the PDADMA/PSS coated catheter, PVC surface charge of the surface plays a dominant role while hydrophobicity and roughness do not play a significant role. These conclusions are also consistent with our previous studies on the influence of poly(allylamine hydrochloride)/sodium poly(4-styrenesulfonate) (PAH/PSS) properties on *Pseudomonas aeruginosa* adhesion capacity [14].

Our findings are also in accordance with other studies dealing with bacteria–surface interactions [29]. For example, Maharubin et al. [30] showed that PVC catheter surfaces could be coated with silver nanoparticles or zinc oxide nanowires as antimicrobial agents in order to prevent bacterial attachment and biofilm formation. The importance of using polyelectrolytes for catheter protection was shown by Yu and coworkers [31]. They applied a water-insoluble polyelectrolyte-surfactant complex, poly(hexamethylene biguanide) hydrochloride-sodium stearate for coating the catheter surface. It was shown that such complex coating successfully prevents generation of the biofilm in the case of *Staphylococcus aureus* and *Escherichia coli* adhesion. Moreover, it should be stated here that such studies could be extended to other types of catheters such as endotracheal tubes [32].

Based on the obtained results, it could be concluded that the main factor affecting the interaction between the bacteria and surfaces is the electrostatics [33]. *E. coli* is negatively charged and that negative charge on bacteria leads to the attractive interaction with positively charged surfaces. The electrostatic interaction between negatively charged *E. coli*

and negatively charged surfaces is repulsive. These mean that *E. coli* has a low adhesion extent to negatively charged surfaces, as indicated in the schematic presentation (Figure 6).

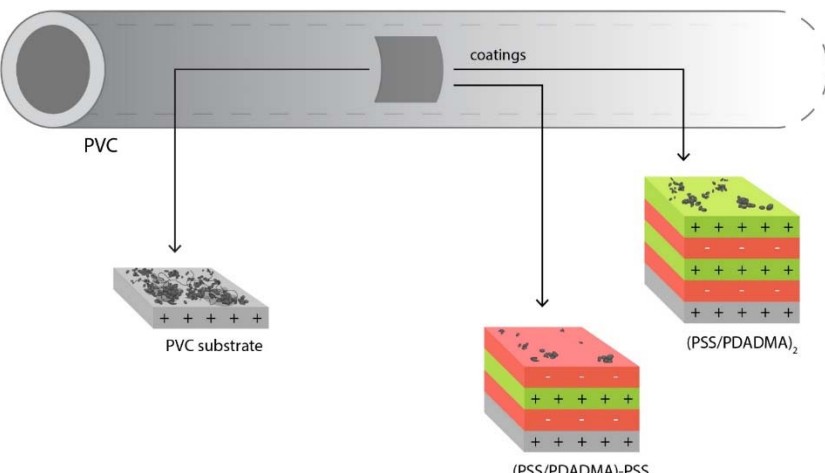

**Figure 6.** Schematic presentation of cylindrical PVC catheters with two types of possible coatings: PSS- and PDADMA-terminated multilayer.

It is important to notice that in Figure 4A, the formation of biofilm, which is defined [34] as a community of bacterial cells irreversibly anchored to a surface inserted in a matrix of extracellular polymeric substances produced by bacterial cells, could be observed. Biofilm represents a very compact structure which is very hard to remove from the surface. In addition, bacterial cells exhibit an altered phenotype concerning growth rate and gene transcription and which makes the problem even more difficult. Therefore, it is very important that biofilms on the catheter surfaces are not formed. This article shows one possibility of the prevention of biofilm formation by the adsorption of PEM coatings on the catheter surfaces.

## 5. Conclusions

In this study, we examined the influence of PVC catheter surface characteristics on the adhesion of uropathogenic *E. coli*. The surface topography was examined by AFM, the zeta potential was determined by electro-kinetic measurements and the contact angle of the materials was measured by tensiometry. From the SEM micrographs, we determined the bacterial adhesion extent. The results demonstrated that surface characteristics and the extent of bacterial adhesion have a positive correlation. This experimental study helps to understand which type of catheter surface coatings can reduce bacterial adhesion when exposed to the urine with uropathogenic *E. coli*. The following main conclusions can be drawn from this study:

- due to its positive potential, the bar PVC catheter surfaces attract *E. coli* very strongly and the biofilm starts to grow,
- the polyelectrolyte coating substantially decreases the bacterial adhesion extent and,
- among polyelectrolytes, the negatively charged polyelectrolytes are due to the strong electrostatic repulsion with negatively charged *E. coli*, the most appropriate surfaces regarding bacterial adhesion.

The results of this study can encourage producers of catheters to improve the surface characteristic and consequently improve the lifestyle of catheter users. The application can be also performed with other types of negatively charged coatings.

**Author Contributions:** Conceptualization, K.B. and D.K.; Data curation, A.A., A.Z., R.Š., and T.K.; Investigation, K.B., D.K., L.K., A.A., A.Z., R.Š., and T.K.; Writing-original draft, K.B. and D.K.; Writing-review & editing, L.K. All authors have read and agreed to the published version of the manuscript.

**Funding:** Funding by Slovenian Research Agency through project J7-2595 (B) and Croatian Science Foundation through project IPS-2020-01-6126.

**Institutional Review Board Statement:** Not applicable.

**Informed Consent Statement:** Not applicable.

**Acknowledgments:** Authors thank the company TIK d.o.o, Kobarid, Slovenia for delivering the catheter surfaces. KB and RŠ thank ARRS through the program "Mechanisms of health maintenance" for financial support. Authors thank M. Krošelj for drawing the scheme.

**Conflicts of Interest:** The authors declare no conflict of interest.

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
