# Peer review of "Bacterial Adhesion Capacity of Uropathogenic Escherichia coli to Polyelectrolyte Multilayer Coated Urinary Catheter Surface"

_coatings, doi:10.3390/coatings11060630_

Round 1

Reviewer 1 Report

1. The material characterisation is well performed with methods that are well established and well documented, however I am missing statisical analysis. Are there any significant difference between the groups? This needs to be done and all figures / tables need to have presented whethere there is sign difference or not.

2. The microbiology part is a little weak. Based on the standard deviation there is no sign difference? Use some other assays too, LIVE/DEAD staining, would be more appriopiate for a biofilm model. 

3. WHy only measuring the E-coli with one type? Maybe compare wild type with a mutant?

4. The bacterial study is done statically, however in a catheter there is flow. I recommend to redo the microbiology in a more relevant model; e.g a flow model

Author Response

We would like to thank the reviewer for valuable comments and encouraging statements about our work with detailed surface characterization.

1. The material characterisation is well performed with methods that are well established and well documented, however I am missing statisical analysis. Are there any significant difference between the groups? This needs to be done and all figures / tables need to have presented whethere there is sign difference or not.

Our reply: The statistical analysis was performed, and the results are included in the manuscript.

2. The microbiology part is a little week. Based on the standard deviation there is no sign difference? Use some other assays too, LIVE/DEAD staining, would be more appropriate for a biofilm model. 

Our reply: Due to dye adsorption into PEM the staining method is not appropriate to study the bacterial adhesion onto PEM. The most reliable method to study adhered bacteria and biofilms onto PEM is SEM.

3. WHy only measuring the E-coli with one type? Maybe compare wild type with a mutant?

Our reply: We used also additional reference strain of positive biofilm producer E. coli ATCC 35218 as a control. With E. coli ACTT 35218 similar observation was obtained as with ACTT 77115.

4. The bacterial study is done statically, however in a catheter there is flow. I recommend to redo the microbiology in a more relevant model; e.g a flow model

Our reply: We made also the study with liquid flow and redid the microbiology. New subsections 2.2.6. and 3.5. were added. The system was described, and the new experimental results were included. 

Reviewer 2 Report

Bohinc et al submit a paper entitled "Bacterial Adhesion Capacity of Uropathogenic Escherichia coli to Polyelectrolyte Multilayer Coated Urinary Catheter Surface".

They evaluate chracteristics of standard polyvinyl chloride (PVC) catheter surfaces and surfaces coated with poly(diallyldimethylammonium chloride)/poly(sodium 4-styrenesulfonate) (PDADMA/PSS) polyelectrolyte multilayers in the application of catheters to the urinary tract (associated with nosocomial infections).

They conclude that the surface charge plays the crucial role in the bacterial adhesion on uncoated and coated PVC catheter surfaces.

1. Can the authors justify why they used a 10^9 CFU/ml bacterial suspension

2. In figure 2, please indicate if the treatments are statistically significant.

3. Same applies for all figures and tables.

4. The discussion lacks comparison with other equivalent studies in the field.

5. The authors talk about biofilm formation, but have no data on cocci which are bactria that produce lots of biofilms. Can they discuss this?

6. Are there relevant studies that used cocci instead of E. Coli?

7. The presence of a recap figure is nie but is a bit too simplistic. indicate the bacteria presence.

Author Response

We would like to thank the reviewer for valuable comments.

1. Can the authors justify why they used a 10^9 CFU/ml bacterial suspension

Our reply: We used an overnight culture were we obtain 109 bacteria/ml, this is the amount of bacteria from we started (International standard ISO7218). Then we diluted this suspension by the ratio 1:300 to obtain 107 bacteria/ml (see subsection 2.2.5.). The urine of a patient with a urinary tract infection contains more than 105 bacteria/ml. Our desire was to get as close as possible to the real situation by patients with urinary tract infection.

2. In figure 2, please indicate if the treatments are statistically significant.

3. Same applies for all figures and tables.

Our reply: The statistical analysis was performed, and the results are included in the manuscript.

4. The discussion lacks comparison with other equivalent studies in the field.

Our reply: At the end of Page 9 and the beginning of Page 10 we added one paragraph about the equivalent studies.

5. The authors talk about biofilm formation, but have no data on cocci which are bactria that produce lots of biofilms. Can they discuss this?

Our reply: Cocci studies related to urinary catheters are less relevant. The most relevant pathogenic species in catheter-associated urinary tract Infection is E.coli in 23.9 %. S. aureus, mostly MRSA ranks last in incidence at 1.6 % (Flores-Mireles A.L. et al, 2019).

References:

Flores-Mireles A.L., Hreha T.N., Hunstad D.A. Pathophysiology, Treatment, and Prevention of Catheter-Associated Urinary Tract Infection. Topics Spin. Cord. Rehab. 2019;25:228–240. doi: 10.1310/sci2503-228. 

6. Are there relevant studies that used cocci instead of E. Coli?

 Our reply: Cocci are less important pathogen. The incidence of S. aureus as catheter-associated urinary tract pathogen is 1.6%. The gram-positive pathogen is not typically considered as a major cause of urinary tract infection. 

7. The presence of a recap figure is nie but is a bit too simplistic. indicate the bacteria presence.

Our reply: The bacterial presence is indicated by black spots.

Round 2

Reviewer 2 Report

changes are ok